# Co-developing a health promotion programme for indigenous youths in Brazil: A concept mapping report

**Paulo T. C. Jardim**[1]*, **Josiliane M. Dias**[1], **Antonio J. Grande**[1], **André B. Veras**[1], **Érika K. Ferri**[1], **Fatima A. A. Quadros**[1], **Clayton Peixoto**[1], **Francielle C. S. Botelho**[1], **Maria I. M. G. Oliveira**[1], **Ieda M. A. V. Dias**[2], **Majella O'Keeffe**[3], **Christelle Elia**[3], **Paola Dazzan**[4], **Ingrid Wolfe**[5], **Seeromanie Harding**[4,5]

**1** Medicine School, State University of Mato Grosso do Sul, Mato Grosso do Sul, Brazil, **2** Federal University of Rio Grande do Sul, Rio Grande do Sul, Brazil, **3** Faculty of Life Sciences & Medicine, Department of Nutritional Sciences, School of Life Course Sciences, King's College London, London, United Kingdom, **4** Department of Psychological Medicine, Institute of Psychiatry, Psychology and Neuroscience, King's College London, United Kingdom, **5** Department of Population Health Sciences, School of Population Health & Environmental Sciences, Faculty of Life Sciences & Medicine, King's College London, London, United Kingdom

* paulo.tacoja@gmail.com

**Data Availability Statement:** All relevant data are within the article and its Supporting Information files.

## Abstract

### Background

Latin America and the Caribbean Region are home to about 42 million Indigenous people, with about 900,000 living in Brazil. The little routinely collected population-level data from Indigenous communities in the region available shows stark inequities in health and well-being. There are 305 Indigenous ethnic groups, speaking 274 languages, spread across the remote national territory, who have endured long-lasting inequities related to poverty, poor health, and limited access to health care. Malnutrition and mental health are key concerns for young people. Building on our Indigenous communities-academic partnerships over the last two decades, we collaborated with young people from the Terena Indigenous ethnic group, village leaders, teachers, parents, and local health practitioners from the Polo Base (community health centres) to obtain their perspectives on important and feasible actions for a youth health promotion programme.

### Methods

The report was conducted in the Tereré Village in Mato Grosso do Sul. Concept mapping, a participatory mixed method approach, was conducted in 7 workshops, 15 adults and 40 youths aged 9–17 years. Art-based concept mapping was used with 9 to 11 years old children (N = 20). Concept systems software was used to create concept maps, which were finalised during the workshops. Focused prompts related to factors that may influence the health and happiness of youths. The participatory method gave Terena youths a significant voice in shaping an agenda that can improve their health.

**Funding:** PTCJ, JMD, AJG, ABV, EKF, FAAQ, CP, FCSB, MIMGO and IMAVD were supported by FUNDECT/CONFAP no 05/2018 – MRC: Health Systems Research Networks 2017.Outorga 010/ 2018. MOK, CE, PD, IW and SH were supported by Medical Research Council MR/R022739/1 AJG was also supported by the Academy of Medical Science, the Newton Fund NIFR7\1004.

**Competing interests:** The authors have declared that no competing interests exist.

**Abbreviations:** LAC, Latin America and the Caribbean; UNDRIP, United Nations Declaration on the Rights of Indigenous Peoples; ICHW, Indigenous Community Health Workers.

## Results

Terena youths identified priority actions that clustered under 'Family', 'School', 'Education', 'Socio-economic circumstances', 'Respect' and 'Sport' in response to protecting happiness; and 'Nutrition pattern', 'Physical activity', 'Local environment', and 'Well-being' in response to having a healthy body. Through the participatory lens of concept mapping, youths articulated the interconnectedness of priority actions across these clusters such that behaviours (e.g. Nutrition pattern, drinking water, physical activity) and aspirations (being able to read, to have a good job) were recognised to be dependent on a wider ecology of factors (e.g. loss of eco-systems, parent-child relationships, student- teacher relationships, parental unemployment). In response to developing youth health, Terena adults suggested priority actions that clustered under 'Relationships', 'Health issues', 'Prevention at Polo Base', 'Access to health care', 'Communication with young people', 'Community life', 'Raising awareness' and 'School support'. Their priorities reflected the need for structural transformative actions (e.g. Polo Base and school staff working together) and for embedding actions to protect Indigenous culture (e.g. integrating their cultural knowledge into training programmes).

## Conclusions

Concept maps of Indigenous youths emphasised the need for a health promotion programme that engages with the structural and social determinants of health to protect their happiness and health, whilst those of adults emphasised the need to address specific health issues through preventative care via a school-Polo Base collaboration. Investment in a co-developed school-Polo-Base health promotion programme, with intersectoral engagement, has potential for making Indigenous health systems responsive to the inequalities of youth health, to yield dividends for healthy ageing trajectories as well as for the health of the next generation.

## Introduction

The Latin America and Caribbean (LAC) region includes many low- and middle-income countries with fragile economic and health systems and is home to over 42 million Indigenous people, representing ~9% of Indigenous peoples globally [1]. Latin America is known to be the most inequitable region in the world and it is unlikely to change its future and achieve sustainable development unless the health of young Indigenous people is also addressed.

There is a general lack of routinely collected population-level data from Indigenous communities in the region, which limits our understanding of the health, wellbeing and development of Indigenous young people. Most reports point to ongoing stark disparities that have persisted despite several legal and constitutional reforms in the LAC region, which were established in response to major international agreements such as the International Labour Organization's Convention No. 169 on Indigenous and Tribal Peoples (1989) and the United Nations Declaration on the Rights of Indigenous Peoples (UNDRIP, 2007). Using data from a range of sources including censuses and country specific surveys and statistical bulletins, the 2015 World Bank estimated that Indigenous peoples make up 8 percent of the population in the LAC region, 14 percent of the poor and 17 percent of the extremely poor [1]. The proportion of Indigenous households living in poverty is twice that of non-Indigenous households living

in poverty, 2.7 times as high for extreme poverty and is three times as high for people living on less than US$1.25 a day [2].

Brazil has about 900,000 Indigenous peoples, with 305 different Indigenous ethnicities speaking 274 languages and spread across remote national territory. They have endured long-lasting inequities related to poverty, poor health and limited access to health care [3]. The socio-economic and health impact of the COVID-19 pandemic on Indigenous communities in Brazil compounds the legacy of inequities [4]. About 42% of Brazil's Indigenous population is <20 years old but this varies geographically, with ~85% in that age group living in the Central-West region [5]. In 2010, more than 70% lived on less than a minimum wage. In the 2010 school Census, 68% of Indigenous students were enrolled in elementary school (7 to 14 years). School infrastructure is poor, with one third of schools located in warehouses, teachers' houses or churches [6]. In 2010, infant mortality among Indigenous populations was almost three times higher than the national average (47.2 versus 16.3/1000 live births, respectively) [7]. The crude death rate was almost three times higher than the national average (15.2 versus 5.8/100000), with 45% of all Indigenous from the 10 to 19 years age group. External causes were the main cause of death in the Indigenous population. Suicide was the main external cause with the highest rates among the 10 to 24 year-olds [8]. Deaths from circulatory and respiratory diseases ranked equal second, reflecting the rise in non-communicable diseases in the Indigenous population. In a national survey on violence against youths (2011 to 2017), only one percent of the sample was Indigenous (n = 3467) but the findings showed higher likelihood of physical (71.8% versus 63.3%) and sexual (29.8% versus 21.3%) violence compared with White Brazilian youths [9]. The first national survey of Indigenous population in 2008–2009 contained 6,707 Indigenous women aged 14 to 49 years and 6,285 Indigenous children aged <5 years in 113 villages across all Brazilian States. Among females a third had anaemia, 46% were overweight or obese and 13% had high blood pressure. Among children the prevalence of stunting, underweight, and wasting were 25.7%, 5.9% and 1.3%, respectively and just over half had anaemia [6].

The State of Mato Grosso do Sul, situated in Central West Brazil, has the second largest Indigenous population in the country. The Terena, an Arawak-speaking people, comprise 36% (~14,000 people) of the Indigenous population in the region [10] and is the focus of the current research. The Terena has a young age distribution, with 48% <21 years old. They live in ocas (Indigenous housing) made of wood and mud, canvas shacks or masonry. Their lands are under constant threat from private farming which puts them in extreme socioeconomic and political precariousness. With the loss of land they would have otherwise cultivated, the Terena seek temporary work as saleswomen on the streets of Campo Grande (capital city of State of Mato Grosso do Sul) or as sugarcane cutters. This proximity to non-Indigenous populations is felt to have influenced their dietary and activity habits and their risk of non-communicable diseases [11]. The prevalence of Type 2 diabetes is ~7% among Terena women compared with 1% among non-Indigenous women [12]. Mental health problems are a major area of concern. There has been a sharp recent increase in violence, associated with substance abuse, mainly among young males [10–12]. As with other Indigenous groups, there are major structural barriers in accessing preventive or curative health care due to sociocultural and linguistic barriers, as well as geographical isolation [10–12]. Some of the Terena villages (e.g. Água Branca, Cachoeirinha) are up to 60 kilometres away from the nearest Polo Base (community health centre). We report on a project conducted with Terena youths, village leaders, teachers, parents, and local health practitioners, which explored their perspectives on important and feasible actions to support the development of a youth health promotion programme.

## Methods

### Ethics statement

Ethical approval for this report was obtained from National Research Ethics Commission (CONEP), protocol CAAE: 89604318.0000.8030, analysis #3.100.358, December 2018 and King's College London Research Ethics HR/DP-20/21-14688. After consultations meetings about the report (see details below), written informed consent was obtained from all participants and additionally for students, written parental consent was obtained. The Village Leader gave written consent for the research team to engage with the school and Polo Base. All consents were witnessed by a member of the research team.

### Setting

The Tereré Village is located on the outskirts of Sidrolândia city in Mato Grosso do Sul, Brazil. The Village has an estimated Terena population of 1,200 inhabitants. There is one school with 300 students registered. Children can be enrolled from the age of 7 years and receive elementary and secondary schooling up to the age of 19 [13]. The school has 5 classrooms, a library and a playroom covered outdoor space for meetings, and a large leisure area with a football pitch. All meetings and concept mapping sessions were held in the school, either in a classroom or in a covered outdoor space.

The leadership structure for the Terena works as: There is a group of community leaders (such as Board members), family representatives and older adults, they have the duty of electing a Village leader and a Vice-leader.

Over the last two decades we have developed an Indigenous community-academic partnership with the recognition that it is vital to respect cultural integrity and agency through the integration of Indigenous knowledge and values [14, 15]. Our team includes researchers of Indigenous ancestries and Indigenous students at the State University of Mato Grosso do Sul, all of whom played a central role in developing these partnerships. For the current report, four targeted consultation meetings were held in 2017 with the Village Leadership, school director and other members of the community to explore ideas and seek formal approval to include the Tereré village in the research proposal that was submitted to funders. Two consultation meetings were held with the village leadership to agree on the focus and to confirm approval, and two meetings were held with other stakeholders (students, teachers, parents, and health practitioners) during which they voiced their concerns about the health of their young people and the urgent need for health promotion programs.

### Concept mapping

Concept mapping methodology uses a participatory approach to combine the tacit knowledge of different groups of stakeholders from different backgrounds. It is also increasingly being used to inform public health interventions and recommended by the World Health Organisation [16–18] as a unique method to embed stakeholder perspectives in intervention development [19, 20]. Concept mapping enables shared understanding and provides '*balance of power'* as the method is predominantly participant-led rather than researcher-driven [21, 22]. There has been a history of exploitative research practices with Indigenous peoples. This has led Indigenous peoples to regard research led by those outside of their communities with distrust. Concept mapping allowed active participation of the Terena youths and adults across all stages of the report and thus the prioritisation of Indigenous values, traditions and knowledge in the concept maps. The method is described in detail elsewhere [17, 23] and follows the steps outlined in Fig 1. Concept mapping activities were conducted with students during four

**Fig 1. The concept mapping process.**

workshops and with adults in three workshops, between January and July 2019. Each concept mapping step was performed in a separate session and lasted about 2 hours.

### Step 1

For concept mapping to capture the views of those who are responsible for the care of young people, it was necessary to have a broad representation from relevant sectors. Over the decade of working with the Terena, researchers were familiar with key stakeholders in the village, including the Village Leadership and those working in primary care and education services. On the advice of teachers and the Village Leader, concept mapping was conducted separately with adults and students. At least 15 participants are recommended for concept mapping [21].

All students enrolled in the school, regardless of literacy levels, were eligible. The total number of students in the school were 300 and 40 were invited based on the advice of teachers who took into consideration who might be available on the day and times that concept mapping sessions took place. Students were broadly representative by gender (45% girls, 55% boys) and age (between the ages of 9 to 17 years old). All sessions with the students were led by the same psychologist (CP) and assisted by 2–3 other members of the research team and Indigenous medical students.

The adult group consisted of 15 Indigenous adults, and comprised of 2 community leaders, 5 elementary school teachers, 2 parents of participating students, 2 Indigenous Community Health Workers, 1 nutritionist and 3 nurses. Indigenous adults were eligible for inclusion if they were from the Tereré Village, had an active role in either village leadership, health service delivery, the school or were parents of students enrolled in the school. All sessions with adults were facilitated by 2–3 members of the research team.

### Step 2

Indigenous students and adults 'brainstormed' statements guided by focused prompts which were previously developed in consultations with Indigenous stakeholders during the approval process. These were: *"To be happy it is necessary to. . ."* and *"To have a healthy body it is necessary to. . ."* for Indigenous students, and *"To develop the health of young people in the village and in school it is necessary that. . ."* for Indigenous adults.

On the advice of teachers, students were subdivided by age as they felt the writing abilities might compromise their contributions. For students aged 9 to 11 years old (n = 20), art-enhanced concept mapping was used to encourage their active participation. Students were invited to a classroom and were provided with paper and colouring pencils. The psychologist used a motivational warm-up session asking for examples of the sort of things that made them happy and sad. The first prompt was introduced after it was clear that all students were comfortable with engaging with the topic. They were given the choice of either writing their statements or drawing their response. Students who chose to draw were invited to then write words/sentences that represented their drawings with the aid of a researcher [24]. The same procedure was used with the second prompt. Students created 27 drawings in response to the focused prompts which resulted in 17 different statements.

Older students, aged 12 to 17 years old (n = 20), were given pencils and paper and asked to write their statements. The statements were then merged to the statements generated by those in the 12 to 17 years age group. The research team then reviewed the statements, removed duplicates and edited them for clarity but not for content as per the standard concept mapping methodology. The process for adults followed that of the older students.

### Step 3

Each statement was written by researchers on individual cards and given to participants. Participants were asked to sort the statements into different categories according to their similarity and to label the categories. Each statement could only belong to one category and grouping all statements in one category was not allowed nor having a category with one statement (13). Following grouping, each adult participant rated the perceived importance and feasibility of change of each statement, relative to each other, on a 5-point Likert scale, ranging from 1 = 'relatively unimportant/feasible' to 5 = 'extremely important/ feasible'. Students rated the statements, relative to each other, on a three-point scale for importance/feasibility (1 = not important/feasible; 2 = important/feasible and 3 = very important/feasible). In a pilot session with the students, responses clustered on a scale from one to three. Other studies have shown this tendency and that a three-point scale can be reliably used for children [25].

### Step 4

Data were manually entered and analysed using the GroupWisdomTM software (Concept Systems, Inc., Ithaca, NY). In summary, each sorting category was first converted to a 0, 1 co-occurrence matrix, that has as many rows and columns as there are statements. It contains 1 in a cell if the row and column statement pair were placed by the participant in the same category and a 0 if the statements were not sorted together. These matrices were then summed across all participants, yielding a similarity matrix that indicated the number of stakeholders that sorted each pair of statements together. Multidimensional scaling analysis used (dis)similarity data and represented them as distances in Euclidean space. Each statement was assigned an 'x' and 'y' value from these analyses which were used in hierarchical cluster analysis to partition the statements into non-overlapping clusters. The statements closer to each other on the map are expected to have been sorted similarly across stakeholders. Clusters are groups of ideas that were grouped together most often and reflect the ideas that were conceptually related according to the participants. All scenarios were then discussed with Indigenous participants before choosing a final number of clusters. A bridging value was computed for each statement and the average for each cluster, which ranged from 0 to 1; the lower the value, the better the coherence from sorting across stakeholders.

### Step 5

Visual representations known as a 'Go-Zone' maps were generated, which provided a visual display of each statement across four quadrants based on ratings of importance and feasibility. The statements in the upper right quadrant contained the statements perceived to be most important and most feasible to change. The maps were used in feedback sessions with the Terena to agree on the interpretation of clusters and on suggestions for a health promotion programme. In addition, a panel of eight Indigenous students from the undergraduate Medical course from State University of Mato Grosso do Sul participated in consultations on the interpretation and implications of the findings.

## Results

### Concept mapping with Indigenous youth

Table 1 shows the synthesis of highly rated statements in the Go Zone [25], based on their response to the two prompts. Drawing examples from 9–11 years old students associated to each theme are represented in Fig 2. S1 Table shows all the statements grouped in their respective clusters. Statements from each cluster are distributed in a map according to their rating

**Table 1. Synthesis of highly rated statements in the Go Zone, based on their response to the two prompts.**

| | Themes (B = average bridging score) | Average Importance Rating* | Average Feasibility Rating* | Statements rated high on importance and feasibility (Go-zone)** |
|---|---|---|---|---|
| **Prompt 1: "To be happy it is necessary to..."** | **Family** (B = 0.05) | 2.65 | 2.70 | 1. Family union; 16. Family care; 18. Improve relationship with mom; 20. Defend family in disputes; 23. Have a good job; 29. Happiness in my family; 40. My father to get a job; 42. Make my family happy; 44. Being with my parents; 46. Peace. |
| | **School** (B = 0.07) | 2.40 | 2.67 | 3. More fun classes; 13. Physical education class; 21. Get good grades; 31. Have friends; 35. Be happy at school; 41. See my teacher happy; 57. Read. |
| | **Education** (B = 0.14) | 2.53 | 2.74 | 5. Be intelligent; 28. To study; 58. Know how to read. |
| | **Socio-economic circumstances** (B = 0.24) | 2.63 | 2.73 | 19. Building a family; 22. Own a house; 30. Joy; 32. Eat tasty foods; 45. My family; 49. Compassion; 59. Have money; 66. Feed me well. |
| | **Respect** (B = 0.30) | 2.42 | 2.48 | 7. Respect among people; 24. Faith; 71. Traveling; |
| | **Sport** (B = 0.60) | 2.31 | 2.41 | 8. Investment in sport; 26. Play sports; |
| | **Average** | 2.49 | 2.62 | |
| **Prompt 2: "To have a healthy body it is necessary to..."** | **Nutrition pattern** (B = 0.06) | 2.60 | 2.61 | 1. Eat fruits; 5. Healthy eating; 12. Drink water |
| | **Physical activity** (B = 0.14) | 2.68 | 2.75 | 6. Sleeping well; 9. Sport at school; 13. Respect for the body; 16. Friendly play; 22. Practical/applied classes; 24. Playing sports; 33. Physical activity. |
| | **Well-being** (B = 0.32) | 2.64 | 2.58 | 15. Have joy and happiness; 17. Behave well; 28. Receive family care; 29. Strong family relationship; 36. Strong community; |
| | **Local environment** (B = 0.54) | 2.58 | 2.50 | N/A |
| | **Average** | 2.63 | 2.61 | |

*Score 1–3

** Derived from the upper right quadrant of the Go-Zone map.

***A list of all the statements derived from each drawing is presented in S1 Table

for importance and feasibility. Statements from each cluster that were rated as most important and feasible are shown in the upper right-hand quadrant of the Map (Fig 3), identified as the "Go Zone". Data obtained through sorting originates Clusters of statements that represent themes of interest and the "go zone"represents the highly rated statements. These results are complementary and, when combined, suggest agreement across participants and which themes and related statements are important for the planning of future interventions [26].

**Prompt 1: "To be happy it is necessary to..."** Seventy-three statements, including five from the drawings, were included in the analysis. Twenty statements were generated from 10 drawings, but 15 statements replicated those of the older age group and were removed (S1 Table).

Six clusters (S1 Fig) were identified: 'Family'; 'School'; 'Education'; 'Socio-economic circumstances'; 'Respect' and 'Sport'. There was a high level of agreement across students in how they classified statements within clusters as reflected by low bridging values (0.05 to 0.30), except for Sport (0.60). Average cluster ratings, derived from the ratings of the individual statements within the cluster, ranged from 2.31 to 2.65 for importance and 2.41 to 2.67 for feasibility. These ranges reflect optimistic ratings for both dimensions on a three-point scale. 'Family' and 'School' had the highest average cluster ratings for importance and 'Education' and 'Socio-economic circumstances' for feasibility. Thirty two of the 73 statements were rated high

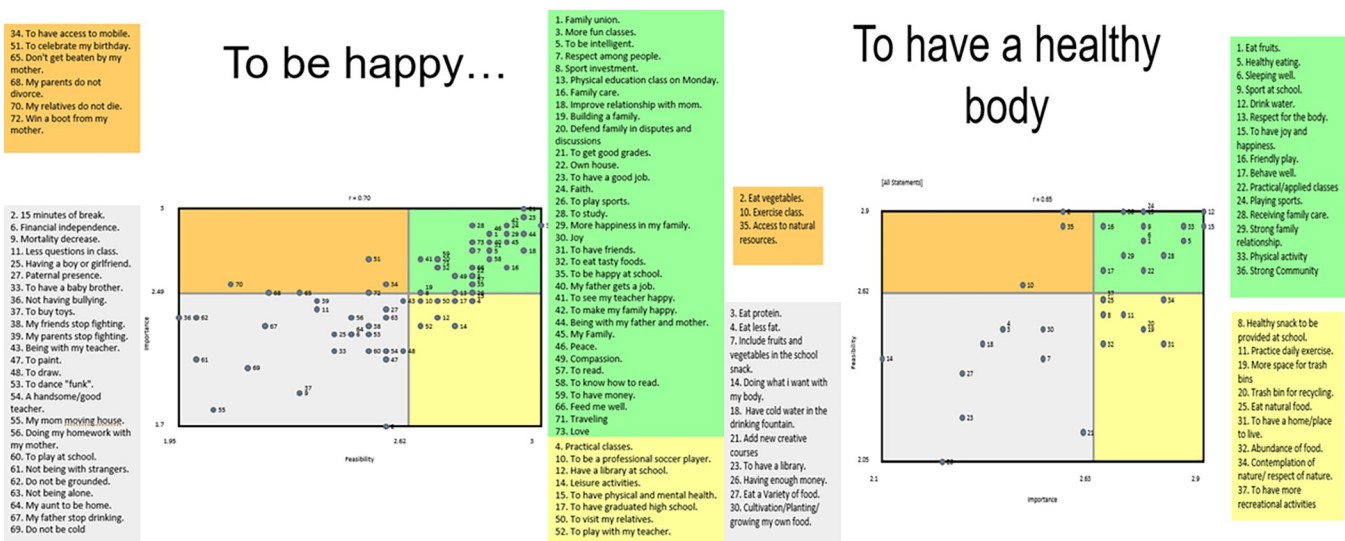

**Fig 2. Drawing examples from 9–11 years old students associated to each theme and based on their perspectives in response to the prompt** *'To be happy it is necessary to...'* **And** *"To have a healthy body it is necessary to...".*

on both importance and feasibility, and these came from three clusters. Perceived priorities were related to family relationships and paternal employment ('Family' cluster); fun learning environment, academic performance, friendships, and physical activity ('School'); and for housing, having money and food ('Socio-economic circumstances') (Table 1). Perceived priorities in the other two clusters related to aspirations to read and study ('Education' cluster), respect ('Respect') and playing sports ('Sports').

The Go-Zone map (Fig 3) displays the average rating scores of all 73 statements across the four quadrants. The average ratings for both importance and feasibility were highest (upper

**Fig 3. Youth Go-zone map based on their perspectives on important and feasible actions in response to the prompt** *'To be happy it is necessary to...'* **And** *"To have a healthy body it is necessary to..."* **Coloured circles indicate to what cluster each statement belongs.**

right-hand quadrant) for having joy, a good job, good grades at school and making their family happy and were lowest for their mothers not leaving the home, not being with strangers and not being cold. The lowest feasibility score was for not being bullied whereas relative high importance, but low feasibility ratings were seen for parents not divorcing and relatives not dying.

*Prompt 2*: *"To have a healthy body it is necessary to. . ."* Thirty-seven statements including 12 from the drawings, were included in the analyses. Eighteen statements were generated from 17 drawings but five statements (e.g. healthy eating, variety of food, playing sports, being strong) replicated those of the older age group and were removed (S1 Table).

Four clusters (S1 Fig) were identified: 'Nutrition pattern'; 'Physical activity'; 'Local environment'; and 'Well-being' (Table 1). Strong agreement in how students classified the statements was evident from low bridging values (0.06 to 0.32) for all but 'Local environment' (0.54). Average cluster ratings ranged from 2.60 to 2.68 for importance and 2.50 to 2.75 for feasibility. 'Physical activity' had the highest average cluster rating for importance and for feasibility, while 'Local environment' had the lowest average cluster rating for importance and feasibility. Twelve of the 37 statements were rated high on importance and feasibility and these were spread across 'Nutrition pattern'; 'Physical activity' and 'Well-being'. Perceived priority actions were related to sleep and sports ('Physical activity' cluster); happiness, family relationships and strong communities ('Well-being') and heathy eating ('Nutrition pattern'). The Go-zone map (Fig 3) displays the average rating scores of all 37 statements. The statements with the highest average ratings for both importance and feasibility included drinking water, having joy and happiness, physical activity, healthy eating, strong family relationships and strong community. Statements that were rated as highly important but not feasible included contemplation of nature, eating natural food, abundance of food and to have a place to live. Low ratings were seen for issues such as having a library, having enough money, cultivating and growing their own foods and having fruits and vegetables in the school snack.

## Concept mapping with Indigenous adults

Table 2 shows the cluster names, their bridging value and average importance and feasibility scores, and for each cluster the statements that were rated as most important and most feasible (upper right-hand quadrant of the Go-Zone map).

*Prompt*: *"To develop the health of young people in the village and at school is necessary to. . ."* Fifty-one statements and eight clusters (S2 Table) were identified: 'Relationships'; 'Health issues'; 'Prevention at Polo Base'; 'Access to health care';'Communication with young people'; 'Community life'; 'Raising awareness'; and 'School support'. Most clusters had low bridging values ($<0.5$) reflecting agreement across students in how they classified the statements. The exceptions were 'School support' (0.50) and 'Community life' (0.68). Average cluster ratings for importance ranged from 4.37 to 4.62 (reflecting high level of importance attributed to all statements) and for feasibility from 3.74 to 4.15 (reflecting those statements were rated relatively feasible to extremely feasible). The 'School support' cluster had the highest average cluster ratings for importance and feasibility.

Twenty one of the 51 statements were rated high on both importance and feasibility, and these were spread across the clusters (S2 Fig). A range of perceived priority actions was suggested. These related to family relationships and working as a team ('Relationships' cluster); improving parent-child dialogue ('Communication with young people'), discussing sexual health, disease prevention, addiction and alcoholism ('Health Issues'), joint working between the school and health teams ('Prevention at the Polo Base'), training teachers and including

**Table 2. Adults: Themes, bridging values, average importance and feasibility ratings and statements for adults based on their response to the prompt: *'To develop the health of young people in the village and at school is necessary to…'*.**

| | Themes (B = average bridging score) | Average Importance Rating* | Average Feasibility Rating* | Statements** |
|---|---|---|---|---|
| *"To develop the health of young people in the village and at school is necessary to…"* | **Relationships** (B = 0.10) | 4.56 | 4.07 | 1. Improve family relationships; 45. Work as a team |
| | **Communication with young people** (B = 0.13) | 4.41 | 3.77 | 24. Improve dialogue between parents and children. |
| | **Health issues** (B = 0.20) | 4.57 | 4.13 | 8. Discuss the importance of disease prevention; 32. Discuss sexual health; 33. Discuss about addiction; 34. Discuss about alcoholism. |
| | **Raising awareness** (B = 0.30) | 4.37 | 3.74 | 43. Raise health awareness. |
| | **Prevention at Polo Base** (B = 0.33) | 4.60 | 4.04 | 7. Conduct prevention events (e.g., day of prevention); 21. Joint work between school and health team; 37. Provide training on the topics covered; 44. Address health care items/priorities; |
| | **Access to health care** (B = 0.46) | 4.52 | 3.96 | 23. Engage people to look for the prevention health unit; 39. Include all health team. |
| | **School support** (B = 0.50) | 4.62 | 4.15 | 20. Include the community in the activity; 38. Commitment of professionals to carrying out the activities; 41. Train teachers. |
| | **Community life** (B = 0.68) | 4.61 | 4.12 | 2. Rescue the culture; 18. Be integrated in the life of the community; 35. Encourage the sport; 36. Encourage leisure activities. |

*Score out of 5.

** Derived from the upper right quadrant of the Go-Zone map.

community in the activities ('School support') and encouraging sports integration in the life of the community as well as rescuing of the culture (Community life').

The Go-Zone map (Fig 4) displays the average rating scores of all 51 statements. The highest average ratings for both importance and feasibility (upper right-hand quadrant) included priority actions such as training of teachers, joint working between schools and the Polo Base, and rescuing the culture. A relatively low feasibility and importance rating was attributed to using the language of young people and low feasibility, but relatively important rating attributed to improving the patient queue at the Polo Base.

## Discussion

### Participative approach in indigenous health promotion

This report explored the perspectives of Terena youths, teachers, parents, village leaders and local health professionals in the priorities and actions needed to guide the development of a health promotion program for youths. Adults and youths engaged with the entire concept mapping process with enthusiasm and respect, with a strong emphasis on the need to address systemic injustices and protection of their culture. Concept maps of youths emphasised the need for a health promotion programme to engage with structural and social determinants of health to protect their happiness and health, whilst those of adults emphasised the need to address specific health issues through preventative care via a school-Polo Base collaboration. We highlight some salient issues relating to Indigenous engagement and participation and the broader determinants of Indigenous youth health and well-being which would be critical for the success of a health promotion programme.

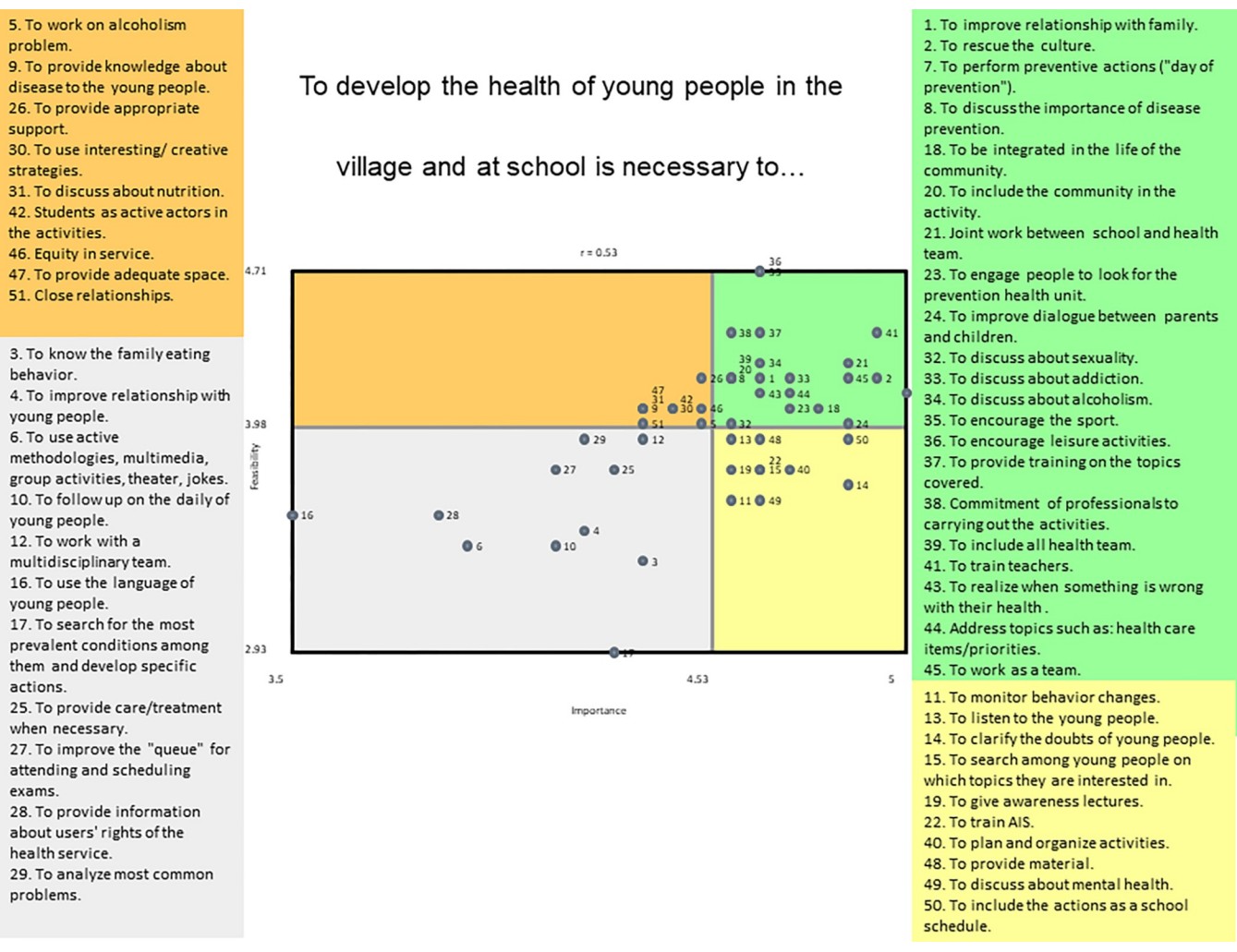

**Fig 4. Adult Go-zone map based on their perspectives on important and feasible actions in response to the prompt '***To develop the health of young people in the village and at school is necessary to***. . .' Coloured circles indicate to what cluster each statement belongs.**

The report is novel in giving Indigenous young people a significant voice in shaping an agenda that can improve their health and aligns with the call by Indigenous youths in the region for effective consultation processes. In a unique meeting in 2017 organized by UNDRIP and the Pan American Health Organization, Indigenous youths from Bolivia, Brazil, Colombia, El Salvador, Guatemala, Guyana, Mexico, Nicaragua, Peru, Panama, Suriname and Uruguay emphasised the need for the creation of opportunities for dialogue and participation in health-related actions [27].

## Youth perspectives on their health

In the current report, Terena youths articulated issues that reflected intuitive recognition of the fundamental social determinants of health. These included socio-cultural factors beyond the individual level, and incorporated an ecology of causes and consequences. They cogently interweaved structural (e.g. parental job loss, owning a house, loss of strong community) and social (e.g. family relationship, happy teachers, friendships) factors and linked these to their health and well-being (e.g. physical activity, healthy eating). Additional poignant points related

to their optimism about aspirations (learning to read, achieving good grades), and their pessimism in addressing issues they felt were important but not achievable such as death of their relatives and eating natural foods. Historically, the Terena people lived in open spaces with land to cultivate, but due to the loss of their lands they now live-in villages in a discontinuous territory, fragmented and surrounded by towns and farms that employ them as temporary workers. Addressing these social and structural determinants of health requires integrated and intersectoral policy actions that tackle the socio-economic-political factors that affect their health.

## A pathway to promote culture and health

Although there was correspondence in ideas in the concept maps of adults and youths, such as the need to address parent-child relationships and encourage sport, there was a nuanced difference in emphasis which related mainly to targeting specific health concerns (e.g. sexual health, addiction, alcoholism, mental health), and enthusiasm for school and Polo Base staff to work together to develop a health promotion programme. The health concerns raised, such as substance abuse and violence among indigenous youths, relate to systemic issues that erode Indigenous self-determination such as territorial dispossession which has led to disrupted family lives and loss of employment [28]. In community-based projects generally, a key challenge is to encourage a culture in which power is shared among adults and youths in building a better society [29, 30]. In the current report, there was a strong commitment from Terena adults, including parents, teachers and health practitioners, to listen to the concerns raised by the Terena youths, which signalled opportunities for bidirectional learning, decision sharing across generations and joint advocacy for change.

## Embedding Indigenous health professionals in the indigenous health care

The provision of education and health services in Indigenous communities in Brazil reflects historic influences from the colonial era, when Catholic priests travelled to the countryside to "teach" religion and provide guidance on a variety of concerns, particularly for young Indigenous people [31]. A recent study with Indigenous nurses, who had acquired their degree in the last 12 months, reported that Indigenous diversity was not covered during their training, and that knowledge construction in their training centred on traditional western medicine and disease conceptualisations. Access to all levels of education among Indigenous people in Brazil is mandated via the National Policy of Education but many argue that the curriculum needs to be culturally centred, incorporate participatory training approaches, and that Indigenous health professionals should be from the village they serve. Currently about 54% of health professional and 10% of teachers are non-Indigenous [32]. It is worth mentioning that in some communities, the transmission of the COVID-19 started with non-indigenous health professionals infected and not tested before entering Indigenous villages [4]. Public health systems are not co-developed with Indigenous communities. A jointly implemented health promotion programme that includes training of Indigenous health practitioners and teachers and which responds to the concerns raised by young people would align with the UNDRIP which calls for Indigenous peoples to have the right to determine their health programmes and to administer these programmes through their own institutions, as well as the right to maintain their traditional health practices.

## The strong relationship between indigenous health and territory

The need for young people to be integrated into community life and to 'rescue the culture' was rated as highly important and feasible actions were included in the adult concept maps. The

youth concept maps showed low feasibility ratings for issues related to their habitats reflecting their pessimism for achieving change. The high importance and feasible ratings, however, for physical activity in both adult and youth concept maps reflected a shared emphasis on the retention of an important aspect of Indigenous life. The Terena have daily long walks to search for food and train to be strong to hunt for birds, monkeys and other mammals and to defend their lands. Sustainability of Indigenous eco-systems is a grave concern. There has been an alarming increase in deforestation rates which has had devastating consequences, due to the spiritual relationship between Indigenous people and the land they inhabit and subsistence economies. About 60% of the Amazon Forest is in the Brazilian territory and is under threat due to fires and environmental degradation caused by cattle rearing, soy production, mining and selective logging [33].

## Strengths and limitations

A participatory approach is crucial in the Indigenous context, unique here in relation to Terena young people. This approach enabled youths to generate their own data in words and pictures, to analyse their data using their own learning styles and to channel their knowledge to shape a health promotion programme for their own health. Arts-based, participatory approaches are widely used in Indigenous contexts and allow youths to voice their concerns about identity, equity and justice [29]. Such innovative methodologies align well with an Indigenous research agenda that resists colonial dynamics and empowers marginalized communities to determine what and how knowledge is generated [34]. Integrating the traditional practice of drawings into the concept mapping process was a powerful method of obtaining the perspectives of the younger age groups as they showed a deep understanding of the issues that affected their health and happiness. The issues they raised also corresponded with the Indigenous specific social determinants that were mentioned by the older age groups, whilst also providing a more granular understanding of the clusters. Concept mapping also illustrated the interconnectedness of issues within and across cluster boundaries, for example by the closeness of the statements on the point maps in the clusters for Family and Socio-economic Circumstances. It captured the variations in Indigenous youth perspectives, including where socio-economic circumstances were perceived to influence their family lives more so for some than others. Ongoing community-academic engagement supported by Indigenous researchers who are co-authors ensured the cultural alignment of methods and feedback on findings from our panel of Indigenous University students enhanced the process of interpretation, particularly into commonalities and differences between the maps of the youths and adults.

Limitations of this report include the inclusion only of those who attended school and the inability to generalise to all Indigenous youths in Brazil as there is much diversity across ethnic groups. Also, youths used a three-point Likert scale and adults a five-point Likert scale to rate the statements for importance and feasibility, which cautions against strict comparability of the youth and adult concept maps.

## Future directions

The issues raised by Terena adults and young people relate to perceived injustices that threaten their habitats, communities, families, schooling and personal aspirations. There are several global (e.g. UNDRIP), regional (e.g Pan American Health Organization Health Plan For Indigenous Youth In Latin America and The Caribbean [35]) and national (e.g. The National Policy on Health Care for Indigenous Peoples in Brazil [36] initiatives that seek to protect the rights of indigenous peoples but there are concerns over the lack of implementation and evaluation.

The following implications of the findings are drawn from the consultations with Terena communities and our panel of Indigenous University students.

The key overarching implications that arise from the findings relate to developing processes that could allow Indigenous young people to advocate for their actionable priorities (protection of their environments, family life, education, employment etc.), capacity building with an intersectoral and participatory lens so that Indigenous communities can implement holistic health promoting strategies that integrate their cultural knowledge, and a unique valuable opportunity for strengthening Indigenous community health systems for the health of young people. Existing initiatives, such as the Health at School Program and ICHWs in each Polo Base, provide a supporting platform from which such an approach to improving community health could be developed. The Terena nurses and ICHWs from the local Polo Base were enthusiastic to co-developing a youth health programme with youths, parents, teachers and Village Leaders. ICHWs are "experience-based experts" who can be drivers for public health action and community empowerment. They have a minimum of elementary schooling, are resident in the community, understand the local geography and culture and navigate across different sectors and services to support patients in the community. Investment in a co-developed school-Polo-Base health promotion programme, with intersectoral engagement, has potential for making Indigenous health systems responsive to the inequities in youth health, to yield dividends for healthy ageing trajectories as well as for the health of the next generation.

## Supporting information

**S1 Table. All statements generated by students in response to prompts 1 and 2, in their respective clusters.**
(DOCX)

**S2 Table. All statements generated by adults in response to prompt To develop the health of young people in the village and at school is necessary to**….
(DOCX)

**S3 Table. Sorting report form statements.**
(XLSX)

**S4 Table. Personal data.**
(XLSX)

**S5 Table. Feasibility and importance data.**
(XLSX)

**S1 Fig.** *Youths' cluster map for prompt 1, 'To be happy it is necessary to*...*' and prompt 2 "To have a healthy body*...*"-***legend: In this map, each point reflects one idea.** Ideas that were grouped together more often appear closer to each other on the map. Ideas never/rarely grouped together appear widely separated on the map. Clusters are groups of ideas that were grouped together most often and reflect ideas that are conceptually related according to the participants in this group. The defined cluster names in this concept maps are: 1- Family, 2- Socioeconomic circumstances, 3- Respect, 4- Sports, 5- School, 6- Education. *Youths' cluster map for prompt 2, 'To have a healthy body it is necessary to*...*'* In this map, each point reflects one idea. Ideas that were grouped together more often appear closer to each other on the map. Ideas never/rarely grouped together appear widely separated on the map. Clusters are groups of ideas that were grouped together most often and reflect ideas that are conceptually related according to the participants in this group. The defined cluster names in this concept maps

are: 1- Nutrition pattern, 2- Physical activity, 3- Well-being, 4- Local environment.
(TIF)

**S2 Fig.** *Adults' cluster map for prompt 'To develop the health of young people in the village and school it is necessary to...' legend* **In this map, each point reflects one idea.** Ideas that were grouped together more often appear closer to each other on the map. Ideas never/rarely grouped together appear widely separated on the map. Clusters are groups of ideas that were grouped together most often and reflect ideas that are conceptually related according to the participants in this group. The defined cluster names in this concept maps are: 1- Relationships, 2- Communication with young people, 3- Health issues, 4- Raising awareness, 5- Prevention at Polo Base, 6- Access to healthcare, 7- School support, 8- Community life.
(TIF)

**S1 Dataset.**
(DOCX)

## Author Contributions

**Conceptualization:** Paulo T. C. Jardim, Josiliane M. Dias, Antonio J. Grande, André B. Veras, Érika K. Ferri, Fatima A. A. Quadros, Clayton Peixoto, Francielle C. S. Botelho, Maria I. M. G. Oliveira, Ieda M. A. V. Dias, Majella O'Keeffe, Christelle Elia, Paola Dazzan, Ingrid Wolfe, Seeromanie Harding.

**Data curation:** Paulo T. C. Jardim, Fatima A. A. Quadros, Majella O'Keeffe, Christelle Elia, Seeromanie Harding.

**Formal analysis:** Paulo T. C. Jardim, Josiliane M. Dias, André B. Veras, Clayton Peixoto, Francielle C. S. Botelho, Majella O'Keeffe, Christelle Elia, Paola Dazzan, Seeromanie Harding.

**Funding acquisition:** Paulo T. C. Jardim, Antonio J. Grande, André B. Veras, Érika K. Ferri, Maria I. M. G. Oliveira, Ieda M. A. V. Dias, Majella O'Keeffe, Christelle Elia, Paola Dazzan, Seeromanie Harding.

**Investigation:** Paulo T. C. Jardim, Maria I. M. G. Oliveira, Majella O'Keeffe, Christelle Elia, Seeromanie Harding.

**Methodology:** Paulo T. C. Jardim, Josiliane M. Dias, Antonio J. Grande, André B. Veras, Clayton Peixoto, Majella O'Keeffe, Christelle Elia, Paola Dazzan, Ingrid Wolfe, Seeromanie Harding.

**Project administration:** Paulo T. C. Jardim, Érika K. Ferri, Fatima A. A. Quadros, Christelle Elia, Seeromanie Harding.

**Resources:** Paulo T. C. Jardim, Francielle C. S. Botelho, Christelle Elia, Seeromanie Harding.

**Software:** Paulo T. C. Jardim, Christelle Elia, Seeromanie Harding.

**Supervision:** Paulo T. C. Jardim, Josiliane M. Dias, André B. Veras, Ieda M. A. V. Dias, Majella O'Keeffe, Christelle Elia, Paola Dazzan, Ingrid Wolfe, Seeromanie Harding.

**Validation:** Paulo T. C. Jardim, Josiliane M. Dias, Fatima A. A. Quadros, Christelle Elia, Paola Dazzan, Ingrid Wolfe, Seeromanie Harding.

**Visualization:** Paulo T. C. Jardim, Josiliane M. Dias, Antonio J. Grande, Christelle Elia, Seeromanie Harding.

**Writing – original draft:** Paulo T. C. Jardim, Josiliane M. Dias, Antonio J. Grande, André B. Veras, Érika K. Ferri, Fatima A. A. Quadros, Clayton Peixoto, Francielle C. S. Botelho, Maria I. M. G. Oliveira, Ieda M. A. V. Dias, Majella O'Keeffe, Christelle Elia, Paola Dazzan, Ingrid Wolfe, Seeromanie Harding.

**Writing – review & editing:** Paulo T. C. Jardim, Josiliane M. Dias, Antonio J. Grande, André B. Veras, Érika K. Ferri, Fatima A. A. Quadros, Clayton Peixoto, Francielle C. S. Botelho, Maria I. M. G. Oliveira, Ieda M. A. V. Dias, Majella O'Keeffe, Christelle Elia, Paola Dazzan, Ingrid Wolfe, Seeromanie Harding.

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
