## [Decision Letter · Decision Letter 0]

4 May 2021

PONE-D-21- 04041

Co-developing a health promotion programme for Indigenous youths in Brazil: a concept mapping study

PLOS ONE

Dear Dr. Jardim,

Thank you for submitting your manuscript to PLOS ONE. After careful consideration, we feel that it has merit but does not fully meet PLOS ONE’s publication criteria as it currently stands. Therefore, we invite you to submit a revised version of the manuscript that addresses the points raised during the review process.

Although this manuscript holds great promise, it must undergo a major revision before it can be resubmitted to PLOS ONE for another review. The most important issues with the manuscript are that – although intrinsically interesting – include two issues raised by the primary reviewer that:

“My primary concern is that the process whereby the results were interpreted is not fully explained. I do not have a sense of where and how this occurred. I do not know how the youth statements were considered, how the decision to advance a school-Polo Base partnership program was made, how the school infrastructure was taken into consideration, which indigenous participants were consulted, what prompts were used in the discussion, who was present, why the university students were brought in, etc. This section needs to be expanded with clear rationale, steps taken, and results outlined.

The discussion and implications sections need to be tethered to the work of this project more specifically. While much of this information is insightful, I need more reflection on the meaning of this work rather than the approach in general or more information on the conditions of the Terena people. Also, it is only in the implications section that we learn specifics about participants and their relationships to the local school system and Polo Base. This should come earlier on in a description of the participants. I would focus more on plans/intentions to enact the recommended program, as it could be considered unethical to generate strong desire without a facilitated pathway forward.”

We look forward to receiving your revised manuscript.

Kind regards,

Kathleen Ragsdale

Academic Editor

PLOS ONE

3. When reporting the results of qualitative research, we suggest consulting the COREQ guidelines: http://intqhc.oxfordjournals.org/content/19/6/349. In this case, please consider including more information on the number of interviewers, their training and characteristics. Moreover, please provide the interview guide used as a Supporting file.

7. We note you have two different versions of Figures 1 and 2 included with your submission, the first embedded in your manuscript (page 10 and page 16) and the second uploaded in the file "Figures (002).docx". So that your figures can be differentiated, can you please update the numbering of the figures and please ensure your figures are uploaded as separate  individual files and not embedded in the manuscript. Further details with regards to figures can be found in the guidelines:  https://journals.plos.org/plosone/s/submission-guidelines#loc-figures-and-tables

8. We note you have multiple versions of your Tables and Supporting Information files uploaded, to avoid confusion can you please ensure you only have a single version of each included. Tables should be embedded in the manuscript, while supporting information should be uploaded as separate supporting information file. Further details with regards to Tables and Supporting Information can be found here: https://journals.plos.org/plosone/s/tables and https://journals.plos.org/plosone/s/submission-guidelines#loc-supporting-information

Reviewers' comments:

Reviewer's Responses to Questions

**Comments to the Author**

1. Is the manuscript technically sound, and do the data support the conclusions?

Reviewer #1: Partly

Reviewer #2: Yes

2. Has the statistical analysis been performed appropriately and rigorously? 

Reviewer #1: Yes

Reviewer #2: N/A

3. Have the authors made all data underlying the findings in their manuscript fully available?

Reviewer #1: No

Reviewer #2: No

4. Is the manuscript presented in an intelligible fashion and written in standard English?

Reviewer #1: Yes

Reviewer #2: Yes

5. Review Comments to the Author

Reviewer #1: The project depicted in this submission is laudable in its aims to address inequities impacting the Terena indigenous population of the Tereré Village in the state of Mato Grosso do Sul, Brazil. The rich description of circumstances facing the Terena people paints a vivid picture of the struggles they face. Overall, the study is well done. However, some deficits in its write-up have led to my suggestion that the paper be revised before acceptance.

More justification up front of why the approach of concept mapping is selected and the researchers’ particular orientation to their work, such as the values and paradigms to which they subscribe, would be helpful.

The description of the concept-mapping workshops was not as revealing as needed. I was left unclear as to the type of facility they were held in, the type of physical set-up used, various roles of researchers in eliciting the information, the exact expectations of participants, how power dynamics were addressed and relationships established, etc. Since the number of youths exceeded methodological recommendations, were they divided into groups beyond asking some to produce artwork? Were they divided in space or time? Were statements by youths written or verbalized? If written, by whom and on what? I do not have a visual depiction of this process.

As for confirmation of the outputs of the software-facilitated analysis, I am left unsure as to which indigenous stakeholders weighed in on the number of clusters, and I am left concerned about their representativeness. I am not an expert on this particular software and cannot evaluate the technical aspects of your computational work, but I find them conceptually sound.

My primary concern is that the process whereby the results were interpreted is not fully explained. I do not have a sense of where and how this occurred. I do not know how the youth statements were considered, how the decision to advance a school-Polo Base partnership program was made, how the school infrastructure was taken into consideration, which indigenous participants were consulted, what prompts were used in the discussion, who was present, why the university students were brought in, etc. This section needs to be expanded with clear rationale, steps taken, and results outlined.

The discussion and implications sections need to be tethered to the work of this project more specifically. While much of this information is insightful, I need more reflection on the meaning of this work rather than the approach in general or more information on the conditions of the Terena people. Also, it is only in the implications section that we learn specifics about participants and their relationships to the local school system and Polo Base. This should come earlier on in a description of the participants. I would focus more on plans/intentions to enact the recommended program, as it could be considered unethical to generate strong desire without a facilitated pathway forward.

There are minor spacing issues in the text, and please note that commas and periods fall within quotation marks and two complete sentences should be separated by periods or semicolons, rather than commas.

The authors state that some restrictions to data access will apply. This certainly makes sense given that qualitative data were collected from participants, some of whom were minors. However, it is not clear from the write-up what types of data would be sensitive, as images produced by the children are depicted. If some parents did not give permission to have their children’s art work shared, or there are other considerations, this should be noted.

To improve future methods, I recommend a more thorough actor mapping process involving easily identified stakeholders to ensure all relevant parties, such as local policymakers, are included in the concept mapping process. Having adequate buy-in, funding, and support for identified solutions among stakeholders at different levels of influence is a key consideration.

Reviewer #2: Overall this is an interesting paper. I do have a few concerns, though.

Introduction - The 1st paragraph in the introduction gives some demographic information about the Indigenous population, but there is no reference to where this information was obtained. This same demographic information is used again in the paper, and it does not have references either. There is also information about youth suicides that does not include the source of the information. Providing sources for these data would strengthen your argument about the concerns for the population you are studying.

Methods - In line 237 you state the purposeful sampling was used. How was the sampling purposeful? What criteria used made the sampling purposeful?

Themes - In Table 1, the themes of Family and Socio-economic circumstances appear to have a lot of overlap. It would be helpful to have a little more information about the data to know why statements were identified under each theme. For example, why is statement 45, "my family" listed under Socio-economic circumstances rather than Family?

6. PLOS authors have the option to publish the peer review history of their article (what does this mean?). If published, this will include your full peer review and any attached files.

Reviewer #1: No

Reviewer #2: No

---

## [Author Response · Author response to Decision Letter 0]

1 Jul 2021

Dear Editor

Please, find attached all of our comments and changes.

---

## [Decision Letter · Decision Letter 1]

9 Dec 2021

PONE-D-21-04041R1

Co-developing a health promotion programme for Indigenous youths in Brazil: a concept mapping study

PLOS ONE

Dear Dr. Jardim,

Thank you for submitting your manuscript to PLOS ONE. After careful consideration, we feel that it has merit but does not fully meet PLOS ONE’s publication criteria as it currently stands. Therefore, we invite you to submit a revised version of the manuscript that addresses the points raised during the review process.

The authors have done an excellent job addressing the first round of reviews and revising the manuscript accordingly. I am pleased to provide a decision of “minor revision” to the revised manuscript. The authors will need to address the second round of reviews from the R1 reviewers extremely carefully – and document their response to each of the reviewers comments – as there are still a number of issues with the writing, mislabeling of tables, etc. These unresolved and/or remaining issues that have been raised by the R1 reviewers must be completely responded to and/or addressed before the manuscript has fully ‘hit the target’ and is ready for publication in *PLoS One. *I look forward to the next round of revisions.

Please submit your revised manuscript on/before January 23, 2022. If you will need more time than this to complete your revisions, please reply to this message or contact the journal office at plosone@plos.org. Please include the following items when submitting your revised manuscript:

We look forward to receiving your revised manuscript.

Kind regards,

Kathleen Ragsdale

Academic Editor

PLOS ONE

Journal Requirements:

Additional Editor Comments (if provided):

The authors have done an excellent job addressing the first round of reviews and revising the manuscript accordingly. I am pleased to provide a decision of “minor revision” to the revised manuscript. The authors will need to address the second round of reviews from the R1 reviewers extremely carefully – and document their response to each of the reviewers comments – as there are still a number of issues with the writing, mislabeling of tables, etc. These unresolved and/or remaining issues that have been raised by the R1 reviewers must be completely responded to and/or addressed before the manuscript has fully ‘hit the target’ and is ready for publication in PLoS One. I look forward to the next round of revisions.

Reviewers' comments:

Reviewer's Responses to Questions

**Comments to the Author**

1. If the authors have adequately addressed your comments raised in a previous round of review and you feel that this manuscript is now acceptable for publication, you may indicate that here to bypass the “Comments to the Author” section, enter your conflict of interest statement in the “Confidential to Editor” section, and submit your "Accept" recommendation.

Reviewer #1: (No Response)

Reviewer #3: All comments have been addressed

2. Is the manuscript technically sound, and do the data support the conclusions?

Reviewer #1: Yes

Reviewer #3: Yes

3. Has the statistical analysis been performed appropriately and rigorously? 

Reviewer #1: Yes

Reviewer #3: N/A

4. Have the authors made all data underlying the findings in their manuscript fully available?

Reviewer #1: Yes

Reviewer #3: Yes

5. Is the manuscript presented in an intelligible fashion and written in standard English?

Reviewer #1: Yes

Reviewer #3: Yes

6. Review Comments to the Author

Reviewer #1: Great job addressing the previously expressed concerns! The following needs to be amended before publication:

o Abstract needs editing (e.g., Latin America, not Latin American)

o The discussion section can still be improved by adding topic sentences to each paragraph to make the point of

each section clearer. Only supporting and concluding statements should follow. All else should be omitted.

o There appears to be content included that pertains to an altogether different paper submission (See

supplementary information COREQ PJ and letter to the editor referencing a study in the Caribbean.

o There are excessive and confusing supplementary files, which took hours to sort out.

Table 2 in the supplementary files is mislabeled and should be Table 1. The text indicates it has ALL

statements, but the table indicates it has examples of statements.

Table 1 in the supplementary files is mislabeled and should be Table 2. There appears to be two of these

tables in the supplementary files, but I am unsure why.

There are two different Figure 1 and Figure 2s in the supplementary files.

It is unclear how Figure 2 in the supplementary files is different from Go-Zone S2-P1.

S2 Fig in the text appears to be labeled Go-Zone S2-P1 in the supplementary files.

The reference in the text to S4 Fig does not appear to correlate with any particular Go-Zone map in the

supplementary files?

The connection between Fig 2 in the text and supplementary files is not clear.

I do not see a S5 Fig in the supplementary files.

I’m generally not making connections between the Go-Zone maps in the supplementary files and the text.

I don’t see references to the following supplementary files in the text and believe they may be superfluous:

S1_Point_Maps; S2 Cluster Maps; and S3 Importance_CRM.

The dataset in S6_dataset has information that is duplicative of what appears elsewhere.

Reviewer #3: The authors excellently responded to the requests/ requirements of the previous reviewer(s). As a first time reader/ reviewers, I was particularly struck by the value of this piece in demonstrating how to give youth voice, alongside their adult counterparts in health promotion practice.

I noted a few things I'd like to mention, which you can adjust as the editor requests-

1) Line 135- Consider using inequitable instead of unequal as the public health fields - and others - are increasingly recognizing that equity, not equality is what we're striving for.

2) Throughout, be sure that you are consistently applying - or not- use of the Oxford comma. It seems that it is sometimes (eg: Line 419) used but not on line 147 or 180, for instance.

3) Line 203- please consider providing more description for the role of the Village Leader. I can make some assumptions about that role, but I don't know for sure who they are -- are they elected, is it honorary, are they an elder, are there multiple leaders, etc?

4) Line 217- please add a citation, website or some additional information to point us to where we can learn about the two-decades long partnership you mention? If it doesn't exist, I believe a full concept paper on this partnership is warranted. We need more of this across many variations of university - community partnerships-- not just with indigenous people. This is a great case study, though.

5) Line 268- please provide a citation for art-enhanced concept mapping or make it more clear that you are providing the only description of that approach that exists in what follows.

6) Line 311- 'Go-Zone' maps-- needs a citation.

7) Table 1- I would highly recommend that you use a different term than "diet". At least in the US culture, that term is increasingly being "sidelined" as it has a connotation for short-term changes to one's nutrition patterns. In this context, you are talking about one's overall nutrition pattern, it seems, so if you could use something like that- "Nutrition pattern"- instead of "diet" that would eliminated an unnecessary possible distraction. (that term appears on line 369 again).

8) Line 434- Early in the paper (eg: abstract) as well as here, I noticed that the phrase "structural and social determinants" was used without a follow-up phrase. It is fairly obvious that you're omitting "of health," but I believe the paper would be enhanced if you would note that either throughout or the first time you mention it (eg: structural and social determinants of health (hereafter referred to as 'structural and social determinants').

9) This is more of a philosophical issue to consider than a firm recommendation- On line 439, you all use the phrase, "The study is..." ... in fact, this is a scholarly report on an ongoing community-academic partnership. The authenticity of the relationship is palpable. I wonder if our continued use of "study" contributes to our continued patriarchal mindset in work of this sort. It is perfectly OK and good to do solid scholarly work like you present here, but I would argue that this is not a "study." It is, instead, a "scholarly report." Or just a "report." As you surely know, words matter and reinforce our ways of thinking. As we try to move away from patriarchal approaches to community engagement and public health, I hope we can shift from using language that reinforces patriarchal thinking.

10) Line- 531- In the implications section, it is somewhat implied with the future directions are, but I had to hunt for it. I suggest that you use a subheading "Future Directions" and complete a paragraph or two on what you plan to do next.

7. PLOS authors have the option to publish the peer review history of their article (what does this mean?). If published, this will include your full peer review and any attached files.

Reviewer #1: No

Reviewer #3: No

---

## [Author Response · Author response to Decision Letter 1]

17 Mar 2022

Reviewer #1

Reviewer #1: Great job addressing the previously expressed concerns! The following needs to be amended before publication:

Thank you

o Abstract needs editing (e.g., Latin America, not Latin American)

Thank you

o The discussion section can still be improved by adding topic sentences to each paragraph to make the point of each section clearer. Only supporting and concluding statements should follow. All else should be omitted.

We agree, we reorganized the discussion with subtopics. It gave fluency to the thoughts developed through the text.However, we believe the text is very concise, thus we did not delete paragraphs.

o There appears to be content included that pertains to an altogether different paper submission (See supplementary information COREQ PJ and letter to the editor referencing a study in the Caribbean.

Thank you. We corrected the title and footnotes. 

o There are excessive and confusing supplementary files, which took hours to sort out.

We agree. We have diminished the number of supplementary files. The idea here was to provide all the data from the study.

♣ Table 2 in the supplementary files is mislabeled and should be Table 1. The text indicates it has ALL statements, but the table indicates it has examples of statements.

Thank you. We fixed the problem.

♣ Table 1 in the supplementary files is mislabeled and should be Table 2. There appears to be two of these tables in the supplementary files, but I am unsure why.

Thank you. We fixed the problem.

♣ There are two different Figure 1 and Figure 2s in the supplementary files.

Thank you. We fixed the problem.

♣ It is unclear how Figure 2 in the supplementary files is different from Go-Zone S2-P1.

Thank you. We fixed the problem.

♣ S2 Fig in the text appears to be labeled Go-Zone S2-P1 in the supplementary files.

Thank you. We fixed the problem.

♣ The reference in the text to S4 Fig does not appear to correlate with any particular Go-Zone map in the supplementary files?

Thank you. We fixed the problem.

♣ The connection between Fig 2 in the text and supplementary files is not clear.

Thank you. We fixed the problem.

♣ I do not see a S5 Fig in the supplementary files.

Thank you. We fixed the problem.

♣ I’m generally not making connections between the Go-Zone maps in the supplementary files and the text.

We explained as bellow:

Statements from each cluster are distributed in a map according to their rating for importance and feasibility. Statements from each cluster that were rated as most important and feasible are shown in the upper right-hand quadrant of the Map (Fig 2), identified as the “Go Zone”. Data obtained through sorting originates Clusters of statements that represent themes of interest and the "go zone"represents the highly rated statements. These results are complementary and, when combined, suggest agreement across participants and which themes and related statements are important for the planning of future interventions. 

♣ I don’t see references to the following supplementary files in the text and believe they may be superfluous:

S1_Point_Maps; S2 Cluster Maps; and S3 Importance_CRM.

Thank you. We fixed the problem.

♣ The dataset in S6_dataset has information that is duplicative of what appears elsewhere.

Thank you. We fixed the problem.

Response to reviewer #3

Reviewer #3: The authors excellently responded to the requests/ requirements of the previous reviewer(s). As a first time reader/ reviewers, I was particularly struck by the value of this piece in demonstrating how to give youth voice, alongside their adult counterparts in health promotion practice.

Thank you for your comments and feedback

I noted a few things I'd like to mention, which you can adjust as the editor requests-

1) Line 135- Consider using inequitable instead of unequal as the public health fields - and others - are increasingly recognizing that equity, not equality is what we're striving for.

We agree, thank you

2) Throughout, be sure that you are consistently applying - or not- use of the Oxford comma. It seems that it is sometimes (eg: Line 419) used but not on line 147 or 180, for instance.

We have removed all Oxford comma

3) Line 203- please consider providing more description for the role of the Village Leader. I can make some assumptions about that role, but I don't know for sure who they are -- are they elected, is it honorary, are they an elder, are there multiple leaders, etc?

The leadership structure for the Terena works as: There is a group of community leaders (such as Board members), family representatives and older adults, they have the duty of electing a Village leader and a Vice-leader

4) Line 217- please add a citation, website or some additional information to point us to where we can learn about the two-decades long partnership you mention? If it doesn't exist, I believe a full concept paper on this partnership is warranted. We need more of this across many variations of university - community partnerships-- not just with indigenous people. This is a great case study, though.

We added two new references, one from the State Government Department and one from the university

5) Line 268- please provide a citation for art-enhanced concept mapping or make it more clear that you are providing the only description of that approach that exists in what follows.

We added new reference: Cooper Y, Zimmerman E. Concept Mapping: A Practical Process for Understanding and Conducting Art Education Research and Practice. Art Education. 2020;73(2):24-32.

6) Line 311- 'Go-Zone' maps-- needs a citation.

We added new reference: Trochim WM, McLinden D. Introduction to a special issue on concept mapping. Eval Program Plann. 2017;60:166-75.

7) Table 1- I would highly recommend that you use a different term than "diet". At least in the US culture, that term is increasingly being "sidelined" as it has a connotation for short-term changes to one's nutrition patterns. In this context, you are talking about one's overall nutrition pattern, it seems, so if you could use something like that- "Nutrition pattern"- instead of "diet" that would eliminated an unnecessary possible distraction. (that term appears on line 369 again).

We agree. Thank you. We changed all over the text

8) Line 434- Early in the paper (eg: abstract) as well as here, I noticed that the phrase "structural and social determinants" was used without a follow-up phrase. It is fairly obvious that you're omitting "of health," but I believe the paper would be enhanced if you would note that either throughout or the first time you mention it (eg: structural and social determinants of health (hereafter referred to as 'structural and social determinants').

We agree. Thank you. We changed all over the text

9) This is more of a philosophical issue to consider than a firm recommendation- On line 439, you all use the phrase, "The study is..." ... in fact, this is a scholarly report on an ongoing community-academic partnership. The authenticity of the relationship is palpable. I wonder if our continued use of "study" contributes to our continued patriarchal mindset in work of this sort. It is perfectly OK and good to do solid scholarly work like you present here, but I would argue that this is not a "study." It is, instead, a "scholarly report." Or just a "report." As you surely know, words matter and reinforce our ways of thinking. As we try to move away from patriarchal approaches to community engagement and public health, I hope we can shift from using language that reinforces patriarchal thinking.

We do agree and we keep consistently all over the text as report

10) Line- 531- In the implications section, it is somewhat implied with the future directions are, but I had to hunt for it. I suggest that you use a subheading "Future Directions" and complete a paragraph or two on what you plan to do next.

We do agree. 

The key overarching implications that arise from the findings relate to developing processes that could allow Indigenous young people to advocate for their actionable priorities (protection of their environments, family life, education, employment etc.), capacity building with an intersectoral and participatory lens so that Indigenous communities can implement holistic health promoting strategies that integrate their cultural knowledge, and a unique valuable opportunity for strengthening Indigenous community health systems for the health of young people. Existing initiatives, such as the Health at School Program and ICHWs in each Polo Base, provide a supporting platform from which such an approach to improving community health could be developed. The Terena nurses and ICHWs from the local Polo Base were enthusiastic to co-developing a youth health programme with youths, parents, teachers and Village Leaders. ICHWs are “experience-based experts” who can be drivers for public health action and community empowerment. They have a minimum of elementary schooling, are resident in the community, understand the local geography and culture and navigate across different sectors and services to support patients in the community. Investment in a co-developed school-Polo-Base health promotion programme, with intersectoral engagement, has potential for making Indigenous health systems responsive to the inequities in youth health, to yield dividends for healthy ageing trajectories as well as for the health of the next generation

---

## [Editor Report · Decision Letter 2]

26 May 2022

Co-developing a health promotion programme for Indigenous youths in Brazil: a concept mapping study

PONE-D-21-04041R2

Dear Dr. Paulo de Tarso Coelho Jardim,

We’re pleased to inform you that your manuscript has been judged scientifically suitable for publication and will be formally accepted for publication once it meets all outstanding technical requirements.

Kind regards,

Akihiro Nishi, M.D., Dr.P.H.

Academic Editor

PLOS ONE

Additional Editor Comments (optional):

The editor believes that the authors adequately addressed the comments and suggestions from reviewers. I am delighted to recommend the manuscript for publication.
---

## [Editor Report · Acceptance letter]

28 Jul 2022

PONE-D-21-04041R2 

Co-developing a health promotion programme for Indigenous youths in Brazil: a concept mapping report 

Dear Dr. Coelho Jardim:

I'm pleased to inform you that your manuscript has been deemed suitable for publication in PLOS ONE. Congratulations! Your manuscript is now with our production department. 

Kind regards, 

on behalf of

Dr. Akihiro Nishi 

Academic Editor

PLOS ONE